# Molecular Characterization of Lactic Acid Bacteria in Bakery and Pastry Starter Ferments

**DOI:** 10.3390/microorganisms11112815

**Published:** 2023-11-20

**Authors:** Jihad Kleib, Ziad Rizk, Abdo Tannouri, Rony Abou-Khalil

**Affiliations:** 1Department of Biology, Faculty of Arts and Sciences, Holy Spirit University of Kaslik, Jounieh P.O. Box 446, Lebanon; jihad.g.kleib@net.usek.edu.lb (J.K.); ziadrizk@usek.edu.lb (Z.R.); abdotannoury@hotmail.com (A.T.); 2Lebanese Agricultural Research Institute (LARI), Fanar Station, Jdeidet El-Metn, Fanar P.O. Box 90-1965, Lebanon

**Keywords:** lactic acid bacteria, fermentation, *Lactobacillus sanfranciscencis*, ITS, RFLP, antagonistic effect

## Abstract

Bread is the oldest and most essential food consumed by humans, with its consumption exceeding nutritional needs and becoming part of cultural habits. Fermentation is an important step in the bread-making process, giving it its rheological, organoleptic, aromatic, and nutritional properties. Lactic acid bacteria and yeasts are both responsible for the fermentation step and part of the natural flour microbiota. In this study, we aimed to characterize LAB in three types of flour, namely, wheat, oat, and rice flour, using conventional phenotypic and biochemical assays and to carry out molecular-biology-based characterization via studying the *rrn* Operon using RFLP of the ITS region and via PCR using species-specific primers. Additionally, the effect of LAB diversity among the three types of flour and their influence on dough characteristics were assessed. Also, we evaluated the antagonistic effects of LAB on two bacterial (*E. coli* and *S. aureus*) and two fungal (*Botrytis* and *Fusarium*) pathogens. This study showed that LAB are not the predominant species in rice flour, while they were predominant in wheat and oat flour. Additionally, *Lactobacillus sanfranciscencis* was found to be the predominant species in wheat flour, while its presence in oat flour was minor. Finally, through their production of soluble substances, LAB exerted antagonistic effects on the four types of pathogenic microorganisms.

## 1. Introduction

Historically, cereal-based product consumption dates back 12,000 years [1], constituting a major component of the human diet, with a 10–20% greater energy contribution than any other dietary component. Among the most-produced cereals, wheat and rice are the most consumed regarding human nutrition, accounting for 55% of total cereal production. A wide variety of breads rely on wheat as a structuring ingredient that has high nutritional value. Besides being an important source of proteins, essential amino acids, vitamins, carbohydrates, minerals, and fiber, wheat contributes to almost 20% of the required daily energy intake [2].

Besides starch, proteins, particularly gluten, make up the two main flour components that are affected by the dynamic process of bread making. These physicochemical, microbiological, and biochemical changes are mainly induced by yeasts and lactic acid bacteria (LAB). The nutritional quality of bread is affected by LAB and yeast metabolic enzymes. This enzymatic activity is responsible for carbohydrate metabolism as well as for protein and amino acid hydrolysis, yielding nitrogen, an essential element for microbial growth [3,4]. Moreover, the type of cereal flour used in baking strongly affects these chemical modifications, giving each type of bread its unique nutritional value, health benefits, and shelf life and distinctive physical properties (texture, crust, and elasticity), taste, and aroma [5].

Alongside their role as a fermentation agent, LAB are considered an important probiotic agent that improves biological functions in the host. Through their immunomodulatory effects, probiotics can help in the treatment and prevention of several diseases, such as inflammatory bowel disorders (IBD) like Crohn’s disease and Ulcerative colitis, two of the most severe emerging health problems in industrialized countries that are considered the most important colon cancer risk factors [6,7]. In addition to their immunomodulatory effect, LAB may have antagonistic activity against other microbiological species. This antagonistic activity is due to the production, by LAB, of antagonistic substances like metabolic end products (H_2_O_2_, lactic and acetic acids, and CO_2_), bacteriocins, and bacteriocin-like molecules, which have long been considered safe and non-toxic for Eukaryotic cells [8,9], thereby reducing the influence of factors that might affect final product quality as well as shelf life.

Nevertheless, bread making has dramatically changed in the last decades, especially in terms of the processing and the materials applied. Commercial bakeries have adapted to these changes and shifted to newer strategies aiming to ensure better food safety and satisfy societal demands as well as improve their productivity [3]. Among these strategies, a better choice of raw materials has proven to be of great importance to achieve the intended purpose; the use of defined strains of microorganisms can be more advantageous and profitable than employing the traditionally used mixture of undefined strains [10]. Eventually, good characterization of fermentation agents (yeasts and LAB) and the determination of their beneficial properties proved to be crucial to achieving a more controlled process.

In this study, entitled “Molecular characterization of lactic acid bacteria in bakery and pastry starter ferments”, we aimed to characterize LAB and yeasts in three types of flour dough, namely, wheat flour, oat flour, and rice flour, using conventional biochemical identification methods as well as bio-molecular identification methods, in addition to assessing the antagonistic activity of these LAB.

## 2. Materials and Methods

### 2.1. Sample Preparation

Three types of commercial flour were chosen (wheat, oat, and rice). Necessary ratios of flour to distilled water were mixed to form a dough. Doughs were then incubated for 72 h at 30 °C to allow flour microorganisms to grow. After incubation, 10 g of each prepared dough was measured and homogenized using a BagMixer Stomacher (Interscience, Paris, France) in 90 mL of sterile tryptone salt solution. Aliquots of the homogenized solution were prepared and used for further evaluation of dough’s properties and for microorganism characterization.

### 2.2. Study of Doughs’ Properties

#### 2.2.1. pH

The pH values of each of the aliquots were measured using a pH meter (edge pH, HANNA instruments, Padua, Italy).

#### 2.2.2. TTA

Total Titratable Acidity for each of the dough solutions was measured. In total, 10 mL of each dough solution was used. Titration using 0.1 N solution of Sodium Hydroxide (NaOH) from 10 mL of aliquot solution was carried out until a pH of 8.5 was reached in the initial solution. Result was expressed as follows: V NaOH mL/N/10/10 g dough.

#### 2.2.3. D/L Lactic Acid Titration

Enzymatic titration was performed using commercial D-Lactic Acid/L-Lactic acid titration kit (R-BIOPHARM, Darmstadt, Germany). A total of 1 mL of each dough solution was centrifuged for 5 min at 10,000 rpm. Supernatant was then extracted, and titration was performed according to manufacturer’s protocol. Absorbances of each lactic acid isomer, produced after enzymatic reaction, were measured using spectrophotometer (SHIMADZU UV-1800, Kyoto, Japan).

Concentration of lactic acid isomer obtained could therefore be calculated according to the general equation for calculating the concentration:(1)$$c=\left(\frac{V\times MW}{\varepsilon \times d\times v\times 100}\right)\Delta A$$
*V* = Final volume (mL).*v* = Sample volume (mL).*d* = Light path (1 cm).*ε* = Extinction coefficient of NADH (at 340 nm = 6.3 L × mmol^−1^ × cm^−1^).*MW* = Molecular weight of the substance to be assayed (in this case, for lactic acid, *MW* = 90.1 g/mol).

### 2.3. LAB Characterization

LAB characterization was performed using traditional phenotypic and biochemical assays, in addition to molecular characterization techniques.

Six-fold serial dilutions were applied, using tryptone salt solution, to each type of initial homogenized dough aliquot to be used for further analysis. Modified De Man, Rogosa, and Sharpe (MRS) (Merck, Darmstadt, Germany) and SourDough Bacteria (SDB) culture media, both with cycloheximide added (Acros organics, Geel, Belgium) for fungi inhibition, were used for LAB studies [11]. MRS agar is a LAB-specific medium providing favorable growth conditions for these bacteria, while SDB is more specific for the culture and isolation of the *Fructilactobacillus sanfranciscensis* species of LAB [12]. Sabouraud Glucose Agar (SGA) (Oxoid Ltd., Dublin, Ireland) culture media with chloramphenicol (Merck, Darmstadt, Germany), for bacterial growth inhibition, was used for yeast studies [11].

#### 2.3.1. Cultures Preparation and Enumeration

At this stage, 10^−5^ and 10^−6^ dilutions of each dough type were used for culture preparation, and 100 µL of each dilution was deposited and spread in petri dishes. MRS and SDB cultures were then incubated at 30 °C for 72 h, while SGA cultures were incubated at 25 °C for 72 h under anaerobic conditions. A colony count was then performed, and results were expressed in colony forming units (CFU).

Furthermore, tests of the ability of the bacterial population to grow under anaerobic and aerobic conditions were evaluated by inoculating 100 µL of the 10^−5^ and 10^−6^ dilutions in MRS broth under aerobic and anaerobic conditions.

After enumeration of LAB, 30 well-isolated colonies corresponding to each flour type (wheat, oat, and rice) were picked from modified MRS agar cultures and replated using a streaking method on modified MRS agar for characterization. Colonies were incubated at 30 °C for 72 h to allow for proliferation.

#### 2.3.2. Phenotypic and Biochemical Characterization

Gram characterization

Gram reaction and Gram-staining techniques were used to identify the gram phenotypes of each colony. Gram-staining technique was performed using Gram kit (Fluka analytical, Buchs, Switzerland) according to the original Gram protocol [13]. Microscopic examination, conducted after Gram-staining procedure, allowed for the examination of bacterial cells and detection of both morphology (cocci or rods) and Gram profiles (Gram+: purple, Gram−: pink).

Gram reaction was performed by applying a 3% KOH solution to a glass slide. A well-isolated colony was then transferred using a sterile loop and mixed with the solution. If gelling was observed after approximately 60 s of colony application, the bacterial colony was considered Gram−; if no gelling was observed, the bacteria was then considered Gram+.

(a)Catalase test

Production of catalase enzyme was tested using slide or drop tests. A drop of H_2_O_2_ was placed onto a slide, followed by addition of sample belonging to a well-isolated colony using a sterile loop. Immediate production of bubbles, signaling dissociation of H_2_O_2_ into water and O_2_, indicated the presence of a catalase-positive bacteria.

(b)Homo/Heterofermentative test

Modified MRS broth was prepared and distributed into screw cap tubes containing inverted Durham tubes (Durham bells, Merck, Darmstadt, Germany). After autoclaving, the Durham bells were filled with media. Using an inoculation loop, a colony was picked and inserted into the tube, followed by incubation at 30 °C for 72 h to enable growth. Air bubbles inside the Durham tube signal heterofermentative bacteria, while the absence of air bubbles signifies the presence of homofermentative bacteria.

#### 2.3.3. Molecular Characterization of LAB

Gram+, Catalase- bacteria, identified in the phenotypic and biochemical parts of the study, were then used in the molecular characterization methods. Colonies that fit the criteria were then picked and cultured in a modified MRS broth and incubated at 30 °C for 72 h.

(a)DNA extraction:

DNA extraction was performed via mechanical extraction using bead beater technology. A total of 1 mL of each culture broth was transferred into sterile Eppendorf tubes. Then, 180 µm and 600 µm glass beads (Merck, Darmstadt, Germany) were added into each tube to allow for shearing of bacterial cell wall and membrane. Tubes were then transferred into BEAD Blaster 24 machine (Benchmark, Sayreville, NJ, USA). Cell lysis was then performed at 18 °C in a 4-cycles protocol, with each cycle consisting of 20 s at a speed of 4 m/s with a 10 s holding step between each cycle.

(b)DNA quantitation and dilution:

Extracted DNA was then quantified using Nanodrop (BioSpec-nano, Kyoto, Japan). A total of 2 µL of each extracted DNA solution was used after vortexing and centrifugation for 3 min at max speed.

Samples were then diluted using Biopure water (Merck, Darmstadt, Germany) in order to obtain a DNA concentration between 20 and 100 ng/µL of DNA per sample. This concentration is essential for optimal PCR conditions. Higher concentrations of DNA can cause PCR inhibition.

(c)Polymerase Chain Reaction (PCR):

RedTaq^®^ ReadyMix™ PCR reaction mix (Merck, Darmstadt, Germany) was used for all PCR reactions as a reaction mix. This mix is a complete PCR reagent that contains all the essential reactants for the PCR in addition to the Taq DNA Polymerase and an inert dye, with the latter serving as an alternative for the loading dye used for gel electrophoresis. All PCR reactions were carried using a Techne Prime Thermal Cycler (Cole-Parmer, Quebec, QC, Canada).

The composition of the PCR master mix is as follows (Table 1).

Three primer sets were acquired from Merck products (Table 2):16S/2-23S/10 primer set: This primer set consists of universal primers that serve to amplify the *rrn* operon region, which helps in genus identification.tRNA^ala^-23S/7 primer set: These primers serve to amplify the tRNA^ala^ DNA sequence of the *rrn* operon, which was later used for RFLP for species differentiation.*Fructilactobacillus sanfranciscensis* species-specific primers: These primers are specific to *Fructilactobacillus sanfranciscensis* species, the most commonly encountered species in cereal products.

The PCR program starts with a 5 min initial denaturation step at 94 °C, followed by a 35-cycle amplification step consisting of the following sub-steps:Denaturation: 1 min at 94 °C.Primer annealing: 1 min at the optimal primer annealing temperature.Extension: 1 min at 72 °C.

The program ends with a 5 min, 72 °C final extension step followed by holding the temperature at 10 °C.

(d)Restriction fragment length polymorphism (RFLP)

Three types of restriction enzymes were used to create an RFLP profile for the amplified fragments using the tRNA^ala^-23S/7 primer set. The restriction reactions for each restriction enzyme were used according to the provider’s manual (Table 3).

(e)Gel electrophoresis

Amplified fragments were then separated using gel electrophoresis. Afterward, 1.5% agarose (Merck, Darmstadt, Germany) gel was prepared in a 1× TBE buffer stained with an ethidium bromide (EtBr) solution. Migration was then performed for 30 min at 100 V using MYGEL mini electrophoresis system (Accuris instruments, Edison, NJ, USA). For RFLP, a 2% gel was used instead of 1.5%, using the same conditions as before. A 100 bp ladder (Qiagen, Hilden, Germany) was used to determine fragments’ sizes. UV transilluminator (VILBER LOURMAT, Marne-la-Vallée, France) was used to detect fluorescence of DNA-EtBr complexes.

#### 2.3.4. LAB Antagonistic Effects

Lactic acid bacteria’s propensity to inhibit the growth of some pathogenic bacteria and fungi was then evaluated using two methods.

(a)Agar well diffusion assay

This method enabled us to evaluate the antimicrobial activity of LAB against 4 microorganisms: *S. aureus*, *E. coli*, *Fusarium oxysporum*, and *Botrytis cinerea*. Pure cultures of the 4 microorganisms were acquired from the Lebanese Agriculture Research Institute (LARI).

Mueller–Hinton (MH) agar media, the most used media type for this assay [14], was prepared and distributed in petri dishes. Colonies from each of the 4 pure cultures were then extracted using a sterile inoculating loop and suspended in distilled water. Using a sterile cotton swab, colony suspensions were spread on the surface of the MH agar, covering the entire surface. Five wells per plate were made using a sterile cork borer and then filled with 100 µL of LAB inoculated in MRS broth at 30 °C for 72 h prior to the assay. Plates were then incubated at 30 °C for 72 h.

(b)Inoculation in medium containing liquid phase from LAB-containing media.

MRS broth was prepared and inoculated with LAB using a sterile inoculating loop followed by incubation at 30 °C for 72 h. Cultures were then removed and centrifuged for three minutes at maximum speed to separate the cellular phase, containing bacterial cells, from the liquid phase, containing metabolites, proteins, and peptides produced by LAB. Supernatant was then collected and added using a sterile 0.22 µm filter syringe to MH agar media and then poured into petri dishes. With the use of a sterile inoculating loop, colonies from *S. aureus*, *E. coli*, *Fusarium*, and *Botrytis* pure cultures were extracted and inoculated at various spots on the agar medium. Plated were then incubated at 30 °C for 72 h.

#### 2.3.5. Statistical Evaluation

All experiments and tests were performed in triplicates; the reported numerical results are the mean values (±standard deviation) of the triplicate analysis.

For LAB and yeast counts, F-test was used, with a *p* value of <0.05. Variances were determined by calculating the square of standard deviation of the enumerations of the 3 types of flour. For LAB, the F-test gave a result of 2.007, with a *p* value of 0.6, and for yeasts, the F-test yielded a value of 4.000, with a *p* value of 0.4, both of which are >0.05, indicating that there were no significant differences in the colony counts between the 3 types of flour.

## 3. Results

### 3.1. Dough’s Specific Properties

The experimental values of pH and TTA of each dough type are summarized in Table 4, while Figure 1 shows the concentrations of lactic acid D and L isomers in wheat, rice, and oat doughs. Wheat flour dough and oat flour dough showed similar proportions of D/L lactic acid, with 4.33 g/L of D (−) lactic acid and 4.91 g/L of L (+) lactic acid for the former and 0.97 g/L of D (−) lactic acid and 0.925 g/L of L (+) lactic acid for the latter, while a considerable difference was seen in the rice flour D/L lactic acid proportions, where the L (+) lactic acid isomer was produced in larger quantities than the D (−) isomer, with 1.55 g/L of L (+) lactic acid produced, while only 0.55 g/L of D (−) lactic acid was produced.

### 3.2. Colony Count

The colony counts on MRS agar media showed a higher number of colonies than the colony count on SDB agar. This indicates the diversity of the bacterial population present. Inoculation on SGA medium also led to a high number of colonies, indicating the presence of high numbers of yeast species (Figure 2); these yeast species will be characterized in an ongoing study at our institution. Incubation under aerobic and anaerobic conditions led to bacterial growth, indicating the presence of facultative anaerobe microorganisms.

### 3.3. Phenotypic and Biochemical Characterization

All the isolated colonies showed a Gram + profile as determined using the Gram staining and Gram reaction techniques. In total, 17 out of the 30 species isolated from wheat flour dough showed a rod-shaped morphology, 12 out of the 30 oat-flour-isolated species were rod-shaped, and 22 out of the 30 rice-isolated species were bacilli. Out of the 30 isolated colonies, 1 wheat colony, 11 oat colonies, and all 30 isolated rice colonies yielded a positive result in the catalase test. These catalase + colonies do not fit the characteristics of LAB, making them non-LAB; they were therefore disregarded from our study and not used for further analysis. Testing for homo- and heterofermentative species using the Durham tube test was applied after the catalase test for the catalase-negative colonies only. In total, 12 out of the 29 isolated wheat colonies were heterofermentative (41%), while most of the oat colonies were homofermentative species, with only 3 out of 19 colonies producing CO_2_ (16%), indicating a heterofermentative pathway.

### 3.4. Molecular Characterization

Amplification using the 16S/2-23S/10 primer set yielded samples with one, two, and three bands. Most samples yielded two fragments with this set of primers (such as colony Wh6), while two colonies, Wh2 and W6, isolated from wheat flour dough showed a three-band profile. Finally, Sample W18, isolated from wheat flour, yielded only one fragment (Figure 3).

Amplification using the tRNAala-23S/7 primer set yielded one fragment for all the samples due to the presence of one tRNAala sequence, except for the Carnobacterium samples, which yielded two fragments due to the presence of two tRNAala sequences.

RFLP using the HindIII enzyme yielded the same profile for all the samples, consisting of two fragments between 300 and 400 bp. No species discrimination using the HindIII restriction enzyme could therefore be attained. In total, 14 samples, 11 from wheat flour isolations (samples W16-Wh5-W21-W12-Wh4-W3-W23-W10-W13-W11-W4) and 3 from oat (samples O20-O12-O17), produced a three-fragment restriction pattern profile with the HinfI restriction enzyme and a five-fragment profile with the TaqI enzyme (Figure 4). All other samples, isolated from wheat and oat flour, gave the same restriction profiles with HinfI and TaqI enzymes, both of which consisted of a three-fragment profile, apart from two wheat samples, W1 and W24, that resulted in two- and four-fragment profiles, respectively, after digestion with HinfI (Figure 5).

### 3.5. LAB Antagonistic Effects

The propensity of LAB to inhibit other microorganisms’ growth was investigated using the well diffusion assay and by inoculating pathogenic microorganisms in media containing supernatant from LAB cultures.

(a)Well diffusion assay.

The well diffusion assay showed an inhibition zone caused by an antagonistic effect of the LAB against the four microorganisms *S. aureus*, *E. coli*, *Fusarium,* and *Botrytis* (Figure 6).

(b)Inoculation in medium containing liquid phase from LAB-containing media.

After preparing MH agar media containing supernatant from the LAB growth medium, inoculation and incubation of the four microorganisms, *S. aureus*, *E. coli*, *Fusarium*, and *Botrytis*, on the prepared medium showed no apparent growth (Figure 7).

## 4. Discussion

### 4.1. Doughs’ Specific Properties

The TTA reflects the effect of organic acids, which are usually weak acids like lactic and acetic acids, on the acidity of food, which can, in turn, affect food’s organoleptic properties and shelf life [15]. The differences in pH and TTA among the three flour types used in this study reflect a difference in the acidic content and origin of every type. This indicates a difference in the microbial communities prevalent in each type. A difference in the predominant LAB species can have a significant effect on these properties according to a study by Bartkiene et al. [16].

Differences in the overall quantities of lactic acid produced and the proportions of lactic acid isomers can be attributed to differences in microbial communities in each flour type. These communities are very much affected by the composition of the environment they live in and the available nutritive substances, in this case, the flour type [17]. Additionally, studies have shown that most LAB strains produce a racemate of D/L lactic acid, whereas fungal strains’ production of lactic acid is limited to the L (+) isomer [18], which can explain the slight difference in this isomer’s concentration in the wheat sample and the big difference from that of the rice sample. This fact can indicate that fungal activity in rice dough is higher, which explains the higher concentration of the L (+) isomer, while both bacterial and fungal activities are present in wheat and oat flour. The results of the titration assay are concordant with the results of the previous assays, concerning pH and TTA, proving that there is a difference in the fermentation agents in the three types of flour.

### 4.2. LAB Characterization

The presence of catalase + colonies proved the diversity of microorganisms present in the flour microbiota. Additionally, the results obtained with rice samples indicate that LAB are not the dominant bacterial species in this type of cereal. This fact can explain the results obtained with the titration of D/L lactic acid, which gave a higher concentration of the L (+) isomer, indicating that yeasts might be the dominant microorganism responsible for fermentation in rice. However, the presence of LAB in rice, even in small quantities, is signaled by the production of the lactic acid D (−) isomer in small quantities. The obtained results from the previous tests indicated the presence of a diversity of microorganisms in the flour microbiota. The next set of tests aimed to identify the LAB species using molecular biology tools.

Most samples yielded two fragments with the 16S/2-23S/10 primer set, indicating their belonging to the Lactobacillus genus of LAB. Two colonies, Wh6 and W6, isolated from wheat flour dough showed a three-band profile, which indicates the presence of *Carnobacterium* sp. A study by Cinta et al. in 2004 differentiated between *Carnobaterium* species according to the length of the ISR fragments they yielded [19]. Based on the mentioned study, the fragments we obtained in the current study belong to either *C. piscicola* or *C. gallinarum*. Sample W18 isolated from wheat flour yielded only one fragment, indicating the presence of other genera belonging to *Weissella*, *Lactoccus*, *Pediococcus*, or *Leuconostoc*. Finally, sample Wh6, which was also isolated from wheat flour, showed a two-fragment profile that was larger in size than the other fragments present in two-band profiles. This can indicate the presence of a genus other than lactobacillus, or the presence of mutations giving rise to another strain. These results show that wheat flour has greater LAB diversity than oat flour, which is concordance with our previous results concerning dough’s specific properties, showing that a more diverse composition in LAB can affect organic acid production, specifically lactic acid.

Enzymatic restriction using HinfI and TaqI of samples W16-Wh5-W21-W12-Wh4-W3-W23-W10-W13-W11-W4 isolated from wheat flour and samples O20-O12-O17 isolated from oat flour yielded profiles concordant with the profiles generated via the enzymatic restriction of *Fructilactobacillus sanfranciscensis*. These results will be confirmed later using *Fr. sanfranciscensis* species specific primers.

The restriction profiles yielded by all the other samples, isolated from wheat (apart from samples W1 and W24) and oat flour after digestion with HinfI and TaqI enzymes, could belong to any of the following species, namely, *Lb. plantarum*, *Lb. paraplantarum*, *Lb. curvatus*, or *Lb. sakei,* according to a study by Rachman et al. [20]. These results must be further investigated using more advanced molecular technologies such as species-specific primers or sequencing.

Samples that generated an RFLP profile similar to the RFLP generated by *Fructibacillus* were then confirmed using species-specific PCR primers. *Fr. sanfranciscensis* proved to be the predominant species in wheat flour, with a percentage of 46% (Figure 8), while its presence in oat cereal was limited to 16% of the overall LAB population. This fact can explain the different results obtained when testing for doughs’ specific properties and proves once more the effect of LAB on differences in different bread types’ properties.

### 4.3. LAB Antagonistic Effects

Both tests, the agar well diffusion and the inoculation in medium containing the liquid phase from LAB cultures, showed inhibition of the growth of the four studied microorganisms. These two tests confirmed that LAB can have antagonistic effects on bacterial and fungal organisms. This inhibitory effect is mainly caused by metabolites and peptides produced by LAB and released into the surrounding environment, such as organic acids, mainly lactic acid, that can reduce pH, which can have a negative effect on microorganisms’ growth. Antimicrobial activity exerted by LAB can also be caused by the production of Bacteriocin and Bacteriocin-Like Inhibitory substances (BLIS), such as Nisin, Lactacin B, and Lactocin [9].

## 5. Conclusions

The findings of this study shed light on the intricate microbial communities residing within wheat, oat, and rice flours, highlighting the co-existence of both bacterial and yeast populations. This co-habitation not only signifies the complex nature of these ecosystems but also underscores their significant influence on the properties of bread dough during fermentation. This study revealed that lactic acid bacteria (LAB), which are often associated with fermentation, are not the dominant bacterial population in rice flour. Instead, wheat flour exhibited a higher diversity of bacterial species, with *Fructilactobacillus sanfranciscensis* emerging as the predominant species in this type of flour. These insights into the specific microbial compositions of different flour types provide valuable knowledge for bakers and food scientists, allowing for more tailored and precise control of fermentation processes to achieve desired dough characteristics. Furthermore, this study highlighted the potential inhibitory effects of LAB on pathogenic bacterial and fungal species. LAB, through the production of metabolic end products such as lactic acid and the release of antimicrobial peptides like bacteriocins and BLIS, can play a crucial role in promoting food safety by suppressing the growth of harmful microorganisms.

This research underscores the importance of understanding the microbial populations present in various flour types and their intricate relationships. Such knowledge can facilitate the development of strategies to optimize bread dough fermentation, enhance food safety, and ultimately improve the quality of baked products.

## Figures and Tables

**Figure 1 microorganisms-11-02815-f001:**
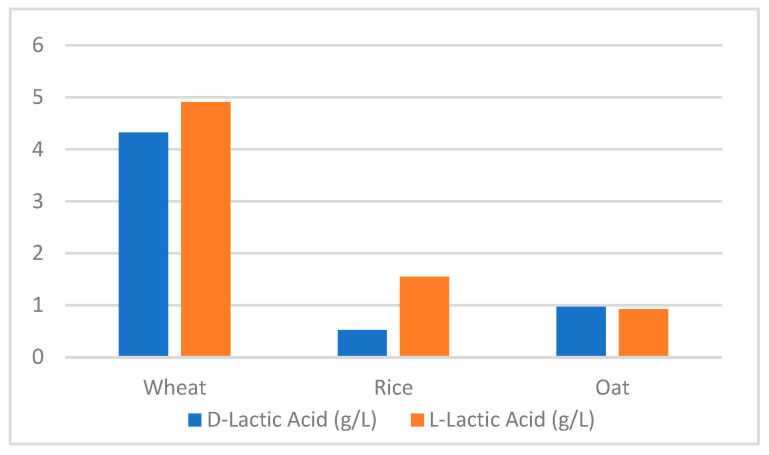
Concentrations of lactic acid D and L isomers in wheat, rice, and oat doughs.

**Figure 2 microorganisms-11-02815-f002:**
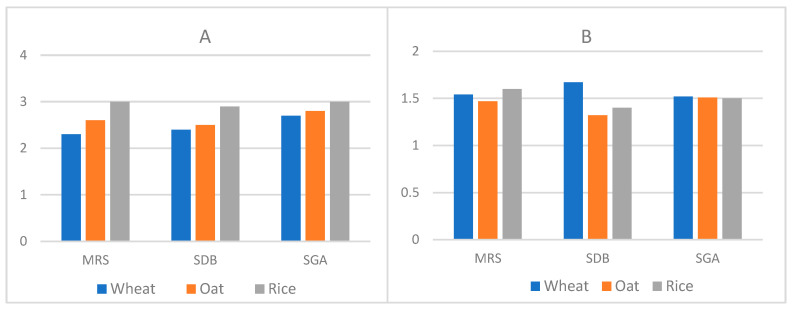
Colony counts. (**A**) LAB colony count (×10^7^ colony), 10^−5^ dilution, and (**B**) yeast count (×10^7^ colony).

**Figure 3 microorganisms-11-02815-f003:**
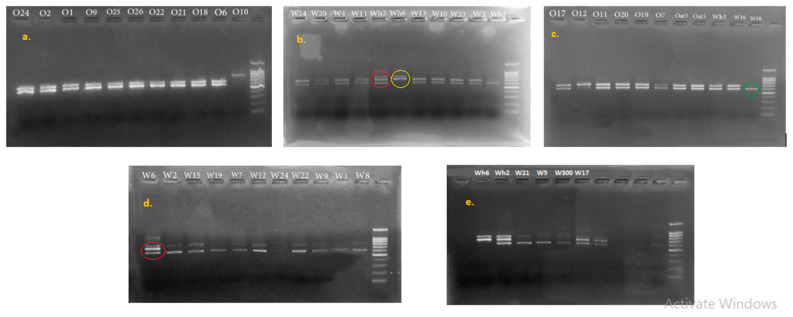
PCR using 16S/2-23S/10 primer set. Colonies isolated from wheat flour dough (**b**–**e**) and oat flour dough (**a**,**c**). Sample W18 yielded only one fragment (**c**; green circle); Sample Wh6 yielded two fragments (**b**; yellow circle); Samples Wh2 and W6 showed a three-band profile (**b**,**d**; red circles).

**Figure 4 microorganisms-11-02815-f004:**
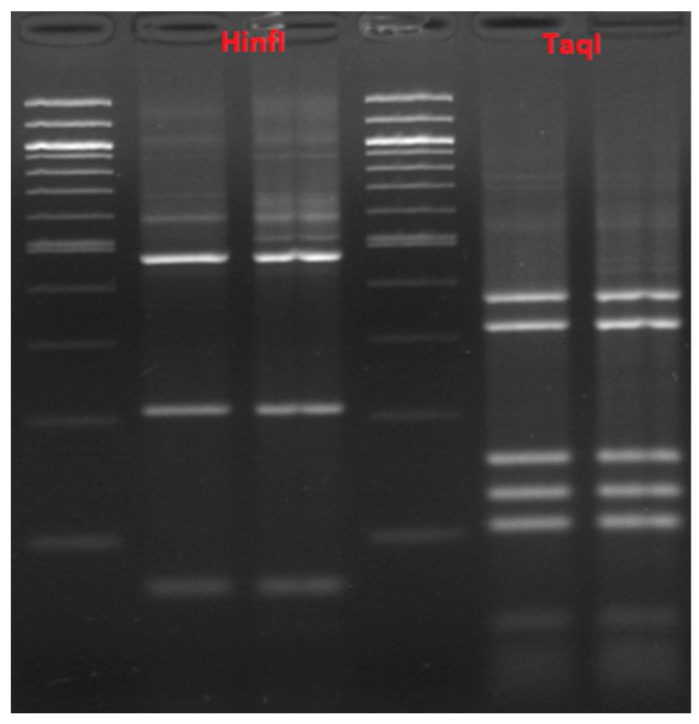
RFLP using HinfI and TaqI of 11 wheat and 3 oat samples.

**Figure 5 microorganisms-11-02815-f005:**
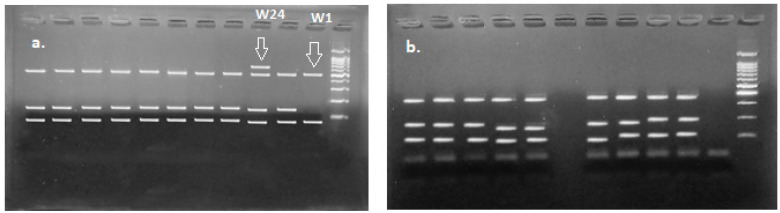
RFLP profiles of the remaining wheat (**a**) and oat (**b**) samples.

**Figure 6 microorganisms-11-02815-f006:**
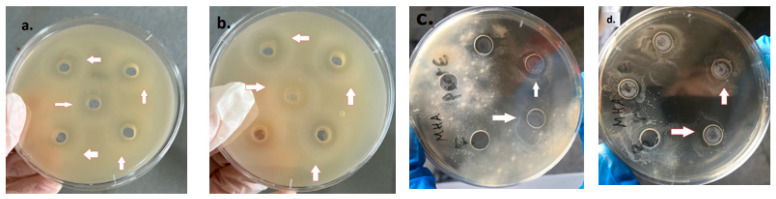
Agar well diffusion assay showing inhibition zones. (**a**): *E. coli*, (**b**): *S. aureus*, (**c**): *Fusarium*, and (**d**): *Botrytis*.

**Figure 7 microorganisms-11-02815-f007:**
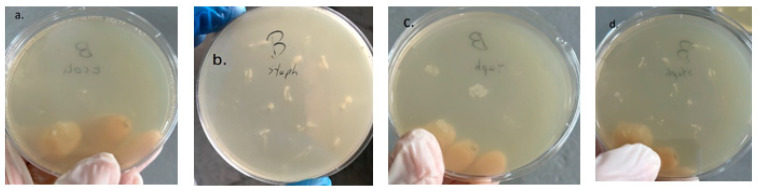
Inoculation in medium containing supernatant from LAB culture: (**a**) *S. aureus*, (**b**) *E. coli*, (**c**) *Fusarium*, and (**d**) *Botrytis*.

**Figure 8 microorganisms-11-02815-f008:**
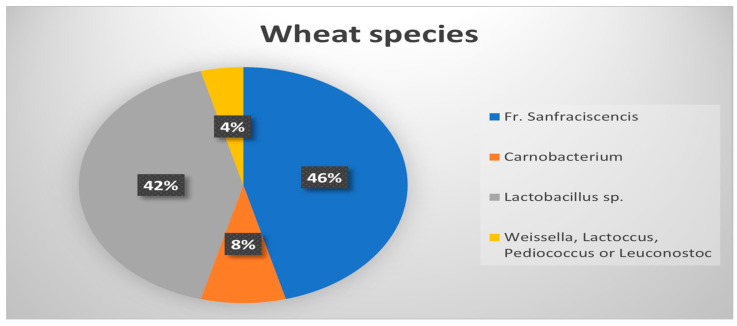
Distribution of species among wheat LAB population.

**Table 1 microorganisms-11-02815-t001:** PCR reaction mix.

Reagent	Quantity (µL)
Reaction buffer mix with MgCl_2_	25
Forward primer (0.5 µM)	1
Reverse primer (0.5 µM)	1
DNA template (20–100 ng/µL)	1
Water (ddH_2_O Nuclease free)	22
Final volume	50

**Table 2 microorganisms-11-02815-t002:** Primer sets.

Primer Set	Forward Primer	Reverse Primer	Optimal Annealing Temperature
16S/2-23S/10	CTT GTA CAC ACC GCC CGT C	CCT TTC CCT CAC GT ACT G	60 °C
tRNA^ala^-23S/7	TAG CTC AGC TGG GAG AGC	GGT ACT TAG ATG TTT CAG	60 °C
*Fructilactobacillus sanfranciscensis* species specific	AAG TCG CCC AAT TGA TTC TTA GT	TTC ACC CTA ATC ATC TGT CCC A	65 °C

**Table 3 microorganisms-11-02815-t003:** Restriction enzymes.

Restriction Enzyme	Source	Restriction Site	Temperature	Provider
HindIII	*Haemophilus Influenzae*	A/AGCTT	37 °C	Thermo Fisher Scientific, Waltham, MA, USA
HinfI	*Haemophilus Influenzae*	G/ANTC	37 °C	Promega, Madison, WI, USA
TaqI	*Thermus Aquaticus*	T/CGA	65 °C	Promega

**Table 4 microorganisms-11-02815-t004:** pH and TTA of the three types of flour.

Flour Type	pH	TTA (mL/N/10/10 g)
Wheat	5.85	5.2
Oat	5.73	4.8
Rice	4.66	4

## Data Availability

The data supporting this research are not publicly available due to privacy considerations.

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
