# Peer review of "Molecular Characterization of Lactic Acid Bacteria in Bakery and Pastry Starter Ferments"

_microorganisms, 2023, doi:10.3390/microorganisms11112815_

Round 1
Reviewer 1 Report
Comments and Suggestions for Authors
The molecular characterization of yeasts were not presented in the manuscript. Why the yeasts species were not characterized. They are very important in bakery.
Announce figures and tables in the text before presenting them. PH must be written pH. The scientific names of bacteria are not well written. Is the bacteria are the dominant microbial population responsible 393
for fermentation in three types of flour, wheat, oat, and rice flours? Review the conclusion because the yeasts were the most counted.
Reviewer 2 Report
Comments and Suggestions for Authors
This paper aimes at the characterization of LAB in three types of flour, the reaearch is valuable to the related researchers, but some aspects need to improve:
1, Tables are all need revision, the format is not suitable;
2,Figure1 and Figure 2 need revise, its not clear and not beautiful;
3,Figure 6 and figure 7 need to rearrange;
4, Conclusions should be 3 or 4 important results.
Comments on the Quality of English LanguageMinor editing of English language required
Reviewer 3 Report
Comments and Suggestions for Authors
The manuscript lacks scientific novelty. The data obtained regarding the colony counts, characteristics of isolated colonies, and RFLP analysis do not provide any information about the bacteria present in the flour, their species composition, or abundance. Currently, approaches based on microbial community profiling using NGS sequencing and quantitative PCR are used for this purpose. In this case, there is a lack of molecular characterization of lactic acid bacteria formed in the starter culture. It seems that the work should be completely revised. By examining how the microbial community changes during fermentation (for example, by comparing the composition of microorganisms in the flour with that formed after fermentation), determining the abundance and compositions of microorganisms, and then describing the dynamics of lactic acid bacteria, a more comprehensive study could be conducted.
Therefore, considering the high standards for publications in microbiology journals, I do not recommend submitting the manuscript for publication in the journal.
Round 2
Reviewer 3 Report
Comments and Suggestions for Authors
Having reviewed the revised version of the manuscript, I still do not recommend it for publication. I believe that the materials received are not enough to publish a publication in a journal at the Microorgansims level. My main comments on the manuscript as a whole. (1) This is the lack of scientific novelty of the results obtained. The presence of lactobacilli in the process of obtaining the test is known, and there are many works on this topic, for example:
1. Alfonzo, A., Ventimiglia, G., Corona, O., Di Gerlando, R., Gaglio, R., Francesca, N., ... & Settanni, L. (2013). Diversity and technological potential of lactic acid bacteria of wheat flours. Food Microbiology, 36(2), 343-354.
2. Scheirlinck, I., Van der Meulen, R., Van Schoor, A., Vancanneyt, M., De Vuyst, L., Vandamme, P., & Huys, G. (2007). Influence of geographical origin and flour type on diversity of lactic acid bacteria in traditional Belgian sourdoughs. Applied and environmental microbiology, 73(19), 6262-6269.
The novelty is due only to the region of production. But in this case, it is necessary to show that the resulting LABs form a separate group among those known at least for the 16S gene. This is not the case in the work and therefore it is not clear what exactly the novelty is at this stage.
The second block of comments is related to the methods used and the conclusions obtained on their basis.
267-272. A larger number of colonies does not indicate the species diversity of bacteria or yeast. This is not true.
278-288. From the materials presented, it is not clear how many total isolates were obtained? And how does this relate to colony count data (pp. 267-272)? The text says “17 out of 30 species isolated from wheat flour” - however, no characteristics of these 30 species are given. It is possible to identify at least 16S rRNA and determine how identical they are. Could this be the same strain? And only two species were sown in 30 colonies?
289-308.
This is a section devoted to the molecular characteristics of the colonies under study. But the problem is that the primers used are already species-specific. This section needs to be significantly supplemented with taxonomic analysis. Comments on the phoresis drawings - the tracks are not labeled, the marker resolution is poor, it is not clear where what sizes are and what sizes were expected?
313-327
Why did you use MRS broth? And not a mixture of isolates? There is no evidence that only LABs grew. Perhaps another type of bacteria had an antagonistic effect?
A general note to all data. There is no information about measurement error everywhere.
198. Who developed the primers used?
Based on the above, I do not recommend this manuscript for publication.
